# Design and Development of a High-Speed Precision Internal Grinding Machine and the Associated Grinding Processes

Zhou Chang [1],*, Qian Jia [2] and Lai Hu [2]

1 School of Mechanical and Electrical Engineering, Lanzhou Jiaotong University, Lanzhou 730070, China
2 State Key Laboratory for Manufacturing System Engineering, Xi'an Jiaotong University, Xi'an 710054, China
* Correspondence: starismyfriend@163.com

**Abstract:** In order to meet the P2-grade bearing grinding requirements, we designed a high-speed internal grinding machine to be used for grinding bearing raceways and inner circles. The machine has a T-type layout and a four-axis numerical control linkage. It is supported by hydrostatic pressure and driven directly by a torque motor. In addition, it is equipped with a high-speed hydrostatic grinding wheel spindle. Our design includes a hydrostatic workpiece shaft and hydrostatic turntable, and the process has a good engineering application value. Finally, the designed precision grinding machine was used to grind a P2-grade bearing raceway.

**Keywords:** internal grinding machine; hydrostatic bearings; high-speed grinding; precision machine design; machining dynamics





## 1. Introduction

A bearing is an important basic machine component whose quality directly reflects a country's level of industrial development. At present, China's basic components cannot meet the national needs, and for many kinds of bearings it is still dependent on imports. Particularly in high-speed and heavy-duty service conditions, the accuracy and load-carrying capacities of Chinese bearings cannot be guaranteed [1]. With the current level of bearing processing in China, it is impossible to process P2-grade rolling bearings effectively in large quantities [2–4].

Grinding is an important processing method for bearing rings, and its importance is self-evident [5,6]. The bearing raceway is the working surface of the bearing, and its processing quality has an important impact on the accuracy, load-carrying capacity and longevity of a bearing. The surface quality of a bearing raceway after grinding is called its surface integrity, and it includes two aspects [7–9]. One is the geometric accuracy of the bearing raceway, usually including the roughness, waviness, roundness, raceway side swing, and groove error. The other is the physical characteristics of the surface of the bearing raceway, which usually include the residual stress, hardness, metallographic structure, and grinding metamorphic layer. The geometric accuracy of a bearing raceway usually affects the bearing's accuracy, while the physical characteristics of the surface of the bearing raceways usually affect the bearing's load-carrying capacity and longevity. At present, there is still a large gap between Chinese and imported bearings in terms of accuracy, load-carrying capacity, and longevity.

Currently, the leading bearing grinder manufacturers are mainly from Europe, Japan, the U.S., and other developed western countries. Heald in the U.S. and TOYO in Japan possess advanced bearing grinder technology. The bearing grinders from these manufacturers can process the outer ring, inner ring, and end face of the bearing with a wide processing range and high processing efficiency and accuracy [10,11].

Due to new trends and developments [12–35], most of these requirements have recently been tackled by improved abrasive processes. The areas of innovation in grinding technology can be summarized as follows:

- Structured grain technology and grain orientation control;
- Modified grinding wheel hub design;
- High-speed grinding (HSG) and high-efficiency deep grinding (HEDG);
- Coolant lubrication technique;
- Hybrid grinding technologies;
- Use of dressings;
- Achieving high surface quality and accuracy.

According to the previous theory and research literature, there is still room for improvement in the P2-grade bearing manufacturing technology. If necessary, the manufacturing technology of P1-grade bearings can be further studied. High surface quality and high accuracy are the research objectives of this study.

## 2. Overall Design Objectives and Technical Indicators of LGID300

For this paper, we designed a high-speed precision grinder, LGID300, to grind P2-grade angular-contact 7014 ball bearings. By using this grinding machine and grinding technology, we sought to achieve precision and load-carrying capacity times of 3000 h for ceramic ball bearings and 2000 h for steel ball bearings. The technical indicators of the 7014 bearings are shown in Table 1. Table 2 shows the comparison of bearing parameters. Table 3 shows the local P2-grade bearing tolerance. Table 4 shows the SKF group P2-grade bearing tolerance.

**Table 1.** Technical indicators of P2-grade angular-contact 7014 ball bearing.

| Bearing | External Diameter | Internal Diameter | Width | Channel/Raceway Roughness |
|---|---|---|---|---|
| B7014 | 110 mm | 70 mm | 20 mm | 0.02 μm |

**Table 2.** Bearing parameter comparison.

| Bearing Code | Overall Dimension | | | | Rated Load | | Limit Speed | | Weight |
|---|---|---|---|---|---|---|---|---|---|
| | Internal Diameter $d$ | External Diameter $D$ | Width $B$ | Contact Angle $\alpha$ | Dynamic State | Static State | Oil | Grease | |
| | mm | | | ° | kN | | r/min | | kg |
| B7014-C-TVP-P4-UL | 70 | 110 | 20 | 15 | 50 | 30.5 | 20,000 | 13,000 | 0.577 |
| 7014C/P4 | 70 | 110 | 20 | 15 | 49.9 | 55.5 | 19,000 | 12,000 | 0.58 |

**Table 3.** Local P2-grade bearing tolerance.

| d/mm | | $\Delta$dmp $\Delta$ds | | $V_{dsp}$ | $V_{dmp}$ | $K_{ia}$ | $S_d$ | $S_{ia}$ [b] | $\Delta_{Bs}$ | | | $V_{Bs}$ |
|---|---|---|---|---|---|---|---|---|---|---|---|---|
| | | | | max | max | max | max | max | | | | max |
| 30 | 50 | 0 | −2.5 | 2.5 | 1.5 | 2.5 | 1.5 | 2.5 | 0 | −120 | −250 | 1.5 |
| 50 | 80 | 0 | −4 | 4 | 2 | 2.5 | 1.5 | 2.5 | 0 | −150 | −250 | 1.5 |

**Table 4.** SKF group P2-grade bearing tolerance.

| d/mm | | $\Delta$ds | | $V_{dp}$ | $V_{dmp}$ | $\Delta_{Bs}$ | | $\Delta_{B1s}$ | | $V_{Bs}$ | $K_{ia}$ | $S_d$ | $S_{ia}$ |
|---|---|---|---|---|---|---|---|---|---|---|---|---|---|
| 30 | 50 | 0 | −2.5 | 1.3 | 1 | 0 | −120 | 0 | −250 | 1.3 | 2.5 | 1.3 | 2.5 |
| 50 | 80 | 0 | −3.8 | 2 | 1.3 | 0 | −150 | 0 | −250 | 1.3 | 2.5 | 1.3 | 2.5 |

The SKF company's bearing manufacturing tolerances and domestic bearing manufacturing tolerances are slightly different. It is worth mentioning that the meanings of the specific parameter values represented in the manufacturing tolerances are also slightly different. One thing is certain, though: the SKF bearing tolerance variance is much smaller than the domestic bearing tolerance variance.

From Table 1, we can see that the accuracy index of the 7014 bearing was high, and its roughness was only 0.02 µm. The bearing raceway needed ultra-precision machining after grinding, and its roughness could reach 0.02 µm. Based on all of the accuracy indices of the P2-grade angular-contact ball bearings, we present the technical indices of the LGID300 ultra-precision internal grinding machine as follows:

1.  Machining accuracy: roundness: 0.3 µm; roughness: 0.3 µm.
2.  Workpiece material: quenched bearing steel, hardness $\leq$ HRC62.
3.  Maximum size of workpiece: inner diameter: 200 mm; width: 160 mm.
4.  Location accuracy: resolution: 0.1 µm; repetitive positioning accuracy: 0.5 µm; absolute positioning accuracy: 3 µm.
5.  X/Z-axis travel: 250 mm; positioning accuracy: 1 µm; repetitive positioning accuracy: 0.3 µm; feed speed: 0.01–4.8 m/min; minimum feed: 20 nm; straightness: whole travel <0.3 µm.
6.  Turntable travel: $\pm 90°$; positioning accuracy: 2.5 arc seconds; repetitive positioning accuracy: 1 arc second; rotational speed: 0.01–10 r/min.
7.  Workpiece shaft: hydrostatic spindle, 1500 r/min; rotation error <0.3 µm.
8.  Grinding wheel axle: maximum speed 50,000 r/min; rotation error less than 0.3 µm.
9.  Grinding wheel dresser: repetitive positioning accuracy: 1 µm.

## 3. Overall Structure Scheme of LGID300

### 3.1. Grinding Scheme and Grinding Wheel Dressing Scheme

The machining method of the inner raceway of the outer ring was cut-in grinding, which can also be called profiling grinding. The grinding wheel was dressed with the rotary dressing tool. The material of the grinding wheel was CBN, and the binder was a ceramic. The grinding wheel granularity is B126/B64, and the grinding wheel concentration is C125/C50. The heat treatment of the grinding wheel's base is T235.

When we ground the inner raceway, we used a diamond roller to dress the formed grinding wheel, which required the workpiece spindle, grinding wheel spindle, and turntable feed to be determined by an interpolation method. The heat treatment of the diamond roller is T235. The sintered corundum layer is D184,C75; D64,C50; D25,C25.

When we ground the inner ring of the bearing, the machine only needed the workpiece spindle and grinding wheel spindle feed to be determined by an interpolation method, and it did not require the rotation interpolation of the turntable.

In order to make full use of the rotational accuracy of the hydrostatic workpiece axis, we made the grinding wheel dresser coaxial with the workpiece axis when we designed the grinding wheel dresser of the LGID300. In this way, we guaranteed the dressing accuracy and reduced the wear of the grinding wheel. In addition, the motor and feed motion of the dresser were omitted, so we could lower the cost and improve the dressing accuracy at the same time. Figure 1 shows the grinding wheel dressing method.

The rotary dressing tool is used in this paper to ensure the effectiveness of the wheel dressing and reduce the loss of grinding wheels during dressing. The rotary dressing tool is used to finish the dressing of the shaped grinding wheel in this paper by interpolating the workpiece axis, the grinding wheel axis, and the rotary table, as shown in Figure 1.

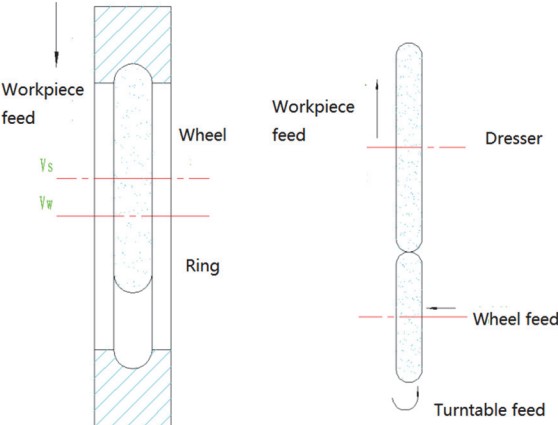

**Figure 1.** Grinding wheel dressing method.

*3.2. Overall Unit Layout of LGID300*

Considering the accuracy of ultra-precision machine tools, it is common to use a multi-axis arrangement with less coupling or uncoupling and to adopt a structure with as small an Abbe error as possible so that the machine structure can achieve a high stiffness and stability. The common structural layouts of ultra-precision machines include overlapping, R-θ, turntable, and T-type layouts. These layouts are suitable for different processing requirements, each having its own advantages and disadvantages [12].

In the design of the LGID300, the maximum diameter of the workpiece was 230 mm, and the width was 160 mm. It was not large, and, thus, the LGID300 can be categorized as a small- or medium-sized machine. According to the above factors, the T-type layout of the LGID300 was selected, which was conducive to improving the rigidity of the machine, strengthening the bearing capacity of the guideways, and guaranteeing the processing accuracy.

The internal grinding design was a fixture clamped on the workpiece axle performing a circular motion. A CBN grinding wheel on the grinding wheel axle performed circular motions at the same time. The workpiece axle worktable drove the workpiece axle to move in the X-direction to perform a linear feed motion. The grinding wheel axle worktable drove the grinding wheel axle to move along the Z-direction to perform a linear feed motion.

The internal grinding required the center axis of the grinding wheel axis to be equal to that of the workpiece axis. We designed a heightening pad to make the two center axes equal. Because the Z-axis linear motion was similar to the X-axis linear motion, we used the same structural design, that is, a hydrostatic guide rail and ball screw, to complete the linear feeding.

In the design of the LGID300, we created a turntable to dress the grinding wheel and mounted the turntable on the Z-axis worktable. We also refined the mechanical structure based on the assembly and operation of the LGID300. Figure 2 shows the grinder layout.

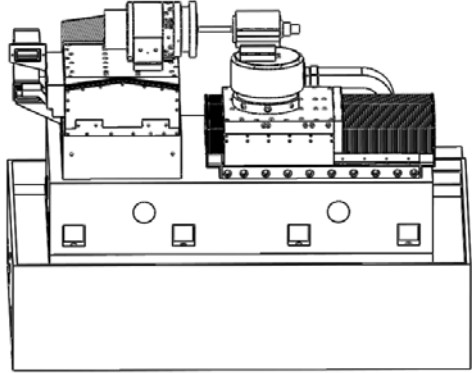

**Figure 2.** Grinder layout.

*3.3. Overall Machine Design of LGID300*

Driving mode: A frameless torque motor drove the mechanical structure directly. About the linear feed axis, the torque motor drove the lead screw directly. About the rotary feed axis, the torque motor drove the rotor directly.

Support mode: The LGID300 was capable of linear motion in the X- and Z-directions and rotary motion of the workpiece spindle and turntable. By adopting a hydrostatic support, we could obtain better motion accuracy and rigidity. Thus, we used hydrostatic guideways in the X- and Z-directions and hydrostatic bearings for the workpiece spindle and turntable.

Precision spindle: We used hydrostatic spindles as the grinding wheel spindle and workpiece spindle. The grinding wheel spindle was purchased, and the workpiece spindle was self-made. The grinding wheel spindle was a 50,000-r/min high-precision hydrostatic spindle produced by the ELKA Company. The hydrostatic spindle was capable of a high rotational speed, high rotational accuracy, and good thermal stability, and it was suitable for the precision grinding of bearings. The motorized spindle had a speed range of 10,000–50,000 r/min, contained a built-in permanent magnet motor, had a maximum sustained power of 8 kW, produced no feedback, and had a 17-mm/M16 tool clamping interface. The radial motion error under all rotation speeds was < 0.06 µm, and the axial motion error under all rotating speeds was < 0.12 µm. The axial stiffness was 69 N/µm, the axial load capacity was 550 N, the radial stiffness was 35 N/µm, and the radial load capacity was 400 N. Figure 3 shows the grinding spindle profile.

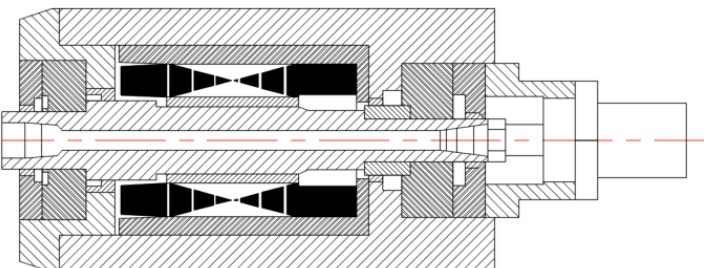

**Figure 3.** Grinding spindle profile.

Overall design: Through the analysis of the grinding process, we selected a reasonable layout for the whole machine, the driving mode, the support mode, and the motorized spindle, and thus we obtained the overall design results for the LGID300. Specific parameters are shown in Table 5. Figure 4 shows a three-dimensional model of the machine tool.

**Table 5.** Main specification parameters of LGID300.

| | | |
|---|---|---|
| Structure Type | | T-type layout and four-axis U-numerical control linkage |
| Outline size Processing object and scope | | 2200 mm × 1700 mm × 1950 mm Workpiece diameter ≤ 230 mm, workpiece width ≤ 160 mm |
| Servo feed system | Driving mode | Torque motor + precision ball screw + hydrostatic guide rail |
| | Axial travel | X axis: 250 mm, Z axis: 250 mm |
| Rotary table C | | Oil static pressure precision turntable, travel ± 90° |
| Workpiece axis B | | Oil static pressure spindle, maximum speed: 2000 r/min |
| Grinding wheel spindle | | Oil static pressure, maximum speed: 50,000 r/min |

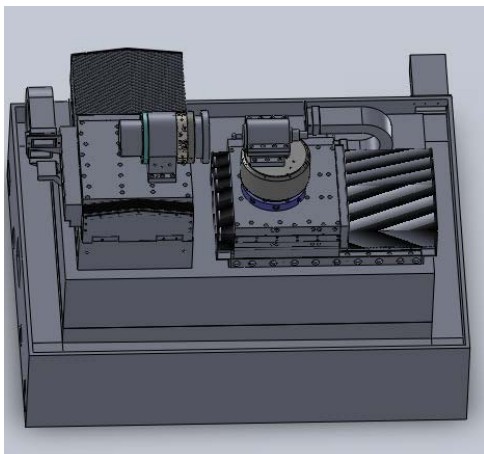

**Figure 4.** Three-dimensional model of machine tool.

## 4. Servo Drive System Design

### 4.1. Screw Selection

We selected the precision ball screw produced by THK in Japan. Considering the cost, we chose C3-grade accuracy. Considering the use space, we chose the nuts and selected the fixed installation mode at both ends. We used a pre-stretching method to assemble the screw and selected angular-contact ball bearings produced by German FAG.

Based on the design indices of the linear feed motion axis and the THK manual, we selected and calculated the guide, diameter, total length, and thread length of the lead screw and completed the stiffness check. The helical pitch of the lead screw was 5 mm, the diameter was 32 mm, the total length of the lead screw was 852 mm, and the thread length was 590 mm. By choosing and designing the screw, we ensured that it met the 250-mm linear travel requirement of the X- and Z-axes.

### 4.2. Selection of Torque Motor and Servo Driver

The friction coefficient of the hydrostatic guideway was very small, with a value of 0.0005. The acceleration design index of the two linear axes of the LGID300 was 0.5. In the process of acceleration and deceleration, the driving moment of the motor mainly needed to overcome the moment of inertia. The rated speed of the torque motor was 3800 r/min, and the continuous stall torque was 4.90 nm. This could meet the requirements for the acceleration, speed, and driving motion of the X- and Z-axes given in the technical specifications. We selected the series frameless brushless motor made by Kollmorgen in the USA to match with the servo driver.

### 4.3. Design of Detection Feedback System

We used an angle encoder to control the speed loop. We selected a British RENIWHAW circular grating. The maximum speed of the grating was 2680 r/min, and the system accuracy was 2.97 arc seconds. After subdivision by a 200-time subdivider, the system could meet the requirements of the positioning accuracy, repetitive positioning accuracy, feed speed, and minimum feed rates for the X and Z linear feed axes.

We used a linear grating ruler to control the position loop. Our selected grating ruler, manufactured by HEIDENHAIN, Germany, is an open grating ruler with high precision and a low price. It is suitable for bearing grinding. The total length of the grating ruler was 270 mm, which met the 250-mm linear travel requirement of the X- and Z-axes. The signal period of the linear grating was 0.512 μm, and the subdivision of the subdivider was 100 times. When the A and B signals were frequency doubled four times in the numerical control system, the positioning resolution of the axes could reach 1.28 nm. This met the requirements of the positioning accuracy and the repetitive positioning accuracy of the X

and Z linear feed axes. For the feed speed, the circular grating and the linear grating also met the requirements.

### 4.4. Results of Servo Drive System Design

In our design, the servo feed mechanism of the X- and Z-axes used a moment motor to drive the lead screw directly through the expansion sleeve. We installed a circular grating at the end of the lead screw and equipped the worktable with a linear grating. We realized the double closed-loop control of the position and speed loop for the feed system to achieve the nano-resolution and sub-micron positioning accuracy of the linear motion axis.

## 5. Design of Hydrostatic Guide Rail

### 5.1. Choice of Guide Form and Throttle

There are two types of hydrostatic guideways: open and closed. The closed hydrostatic guideway had better rigidity than the open guideway. There are two modes of oil supply: quantitative mode and constant-pressure mode. The cost of the constant-pressure oil supply is lower. We selected a liquid closed hydrostatic guideway and a constant-pressure oil supply mode.

We used a new type of gap restrictor in the hydrostatic guide rail and hydrostatic bearing of the LGID300 [13]. Its effective gap size was an annular width $d_c$ and an annular length $l_c$. The structure of the gap restrictor is shown in Figure 5, in which the 1-indenter was used to fix the throttle, and a 2-'O'-shaped sealing ring was used to prevent oil backflow.

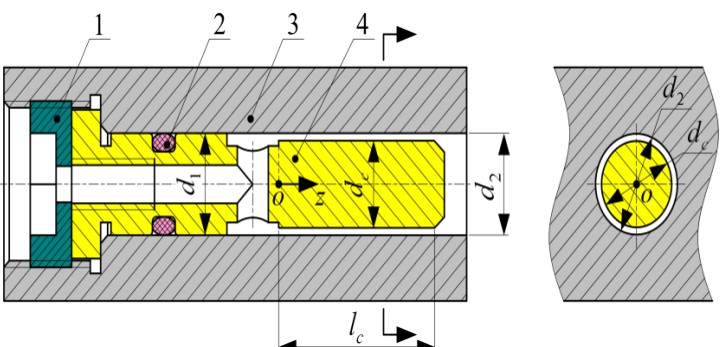

**Figure 5.** Gap restrictor.

Throttling principle: The throttling ratio of the throttle is adjusted by changing the values of dc and lc.

### 5.2. Bearing Capacity and Oil Film Stiffness

The total weights of all parts of the X and Z linear feed axle guides were about 3000 and 3300 N, respectively. The vertical and horizontal static stiffnesses of the hydrostatic guide were calculated to be about 7000 and 5000 N/μm, respectively. The bearing capacity and oil film stiffness met the design requirements.

### 5.3. Guideway Accuracy and Oil Film Thickness

The two relative moving surfaces of the hydrostatic guideway were in a pure liquid friction state, and the oil film stiffness was large, which raised the requirements for the geometric accuracy of the guideway [14]. The typical requirement is

$$\Delta \leq (\frac{1}{2} \sim \frac{1}{3})h_0$$

where $\Delta$ is the total geometric accuracy error of the surface of the guide rail, and $h_0$ is the oil film thickness of the guide rail. The oil film thickness design is derived from the above inequality.

The oil film thickness of the guide rail should not be too large to avoid reducing the rigidity of the guide rail, and it should also not be too small. Generally, the following values are recommended for the design of hydrostatic guideways: the oil film thickness of hydrostatic guideways for medium- and small-machine tools should be 15–30 μm, and that for large-machine tools should be 30–60 μm. Thus, we selected $h_0 = 20$ μm.

### 5.4. Design Results of Guide Rail Structure

In order to make the oil film uniform, we designed two oil chambers. The hydrostatic guide was composed of left and right guide bars and a slide saddle. The sliders moved together with the workbench. We added an oil circuit and an oil-saving hole on the moving sliders. We selected the guide dimensions of B = 40 mm, b = 15 mm, L = 245 mm, l = 20 mm, and h = 20 μm.

## 6. Design of Hydrostatic Rotary Parts

### 6.1. General Design of Hydrostatic Bearing

We selected an overall design scheme with a direct torque motor driving, hydrostatic bearing support, non-contact circular grating feedback, and computer digital control for the hydrostatic turntable and spindle. Table 6 shows the design indices of the hydrostatic bearing.

**Table 6.** Design indices of hydrostatic bearing.

| Items | Hydrostatic Turntable | Hydrostatic Spindle |
|---|---|---|
| Radial motion error | 0.3 μm | 0.3 μm |
| Axial motion error | 0.3 μm | 0.3 μm |
| Positioning accuracy | 2.5 arc seconds | 2.5 arc seconds |
| Repetitive positioning accuracy | 1 arc second | 1 arc second |
| Radial stiffness | 2000 N/μm | 1000 N/μm |
| Axial stiffness | 2000 N/μm | 1000 N/μm |
| Design speed | 100 r/min | 1500 r/min |

### 6.2. Structural Design of Hydrostatic Bearing

There are two types of hydrostatic bearings: radial hydrostatic bearings and bi-directional thrust hydrostatic bearings. Radial hydrostatic bearings mainly bear radial loads, and bi-directional thrust hydrostatic bearings mainly bear axial loads. We selected a structure where the axle was wrapped with a bearing bushing to improve the axial stiffness of the rotating spindle. The hydrostatic bearings used the same throttle as the hydrostatic guide.

### 6.3. Torque Motor and Grating Ruler

The torque motor of the hydrostatic turntable was a KBMS-57 × 01-A. The torque motor of the hydrostatic spindle was a KBMS-43 × 03-A. The parameters of the torque motors are shown in Table 7.

**Table 7.** Technical parameters of torque motor.

| Items | KBMS-43 × 03-A | KBMS-57 × 01-A |
|---|---|---|
| Rated power | 2.52 kW | 2.31 KW |
| Rated torque | 21 N·m | 18.8 N·m |
| Rated speed | 1500 r/min | 2050 r/min |
| Rotor inner diameter | 76.28 mm | 104.17 mm |
| Rotor outer diameter | 159.28 mm | 202.90 mm |

We selected Renishaw's non-contact circular grating for both the hydrostatic turntable and the spindle. It had a nominal inner diameter of 80 mm and an outer diameter of 100 mm, and its number of dividing lines was 15,744. With a 200-time subdivider, the

resolution could reach 0.41 arc seconds. Because the motion of the hydrostatic bearing was smooth, and it showed no creeping phenomenon at low speeds, we could guarantee a positioning accuracy of 2.5 μm and repetitive positioning accuracy of 1 μm by the control of the actuator.

## 7. Accuracy Detection

We measured the radial and axial runout errors of the hydrostatic turntable at a constant temperature of 20 ± 2 °C. The radial and axial runouts of the hydrostatic turntable were 0.31 and 0.28 μm, respectively. The radial and axial runouts of the hydrostatic spindle were 0.32 and 0.28 μm, respectively. Figure 6 shows the turntable rotation accuracy detection. Figure 7 shows the spindle rotation accuracy detection.

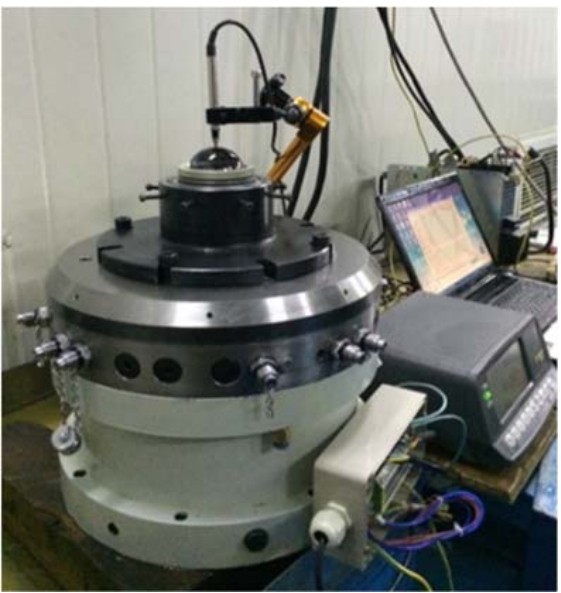

**Figure 6.** Turntable rotation accuracy detection.

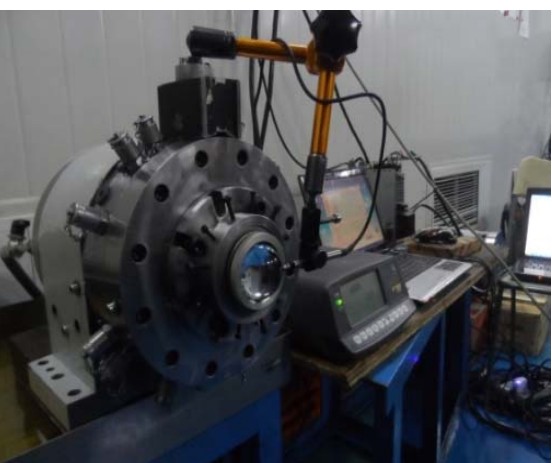

**Figure 7.** Spindle rotation accuracy detection.

Figure 8 shows the X-axis and Z-axis straightness, positioning accuracy, and repeated positioning accuracy test.

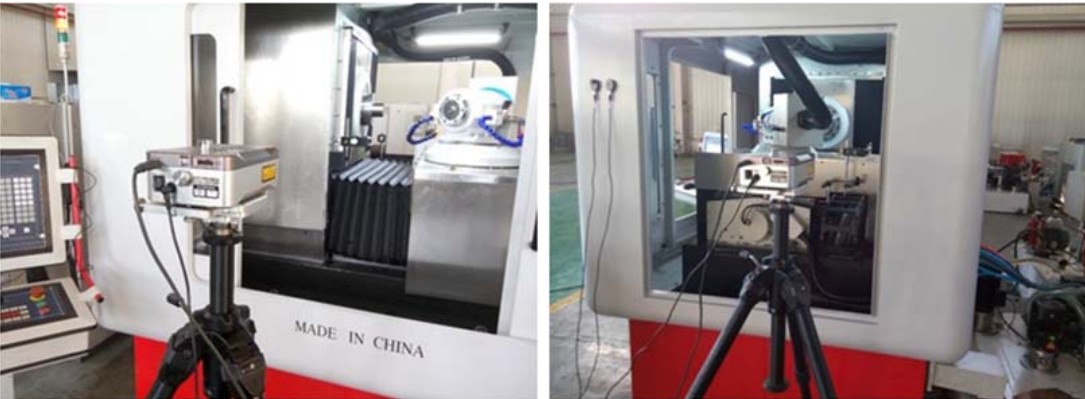

**Figure 8.** X-axis and Z-axis straightness, positioning accuracy, and repeated positioning accuracy test.

## 8. Grinding Test

Figure 9 show that the grinding of the bearing raceway and the grinding process itself were reliable and effective. The use of an adhesive fixture was a major feature of this study. Figure 10 shows the grinding-affected layer. The measuring equipment was a roundness meter. All of the comparison test results are shown in Table 8.

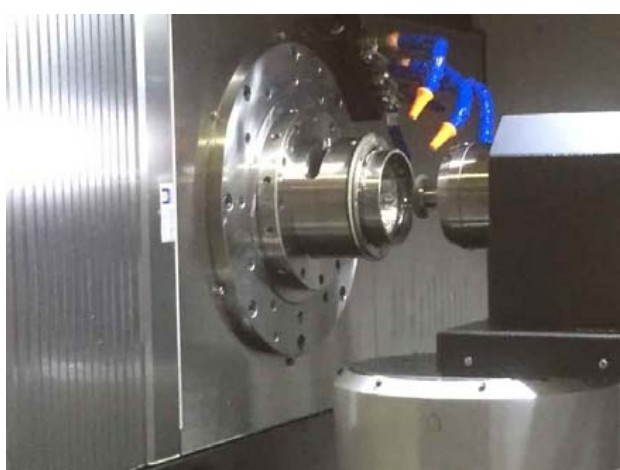

**Figure 9.** Grinding process.

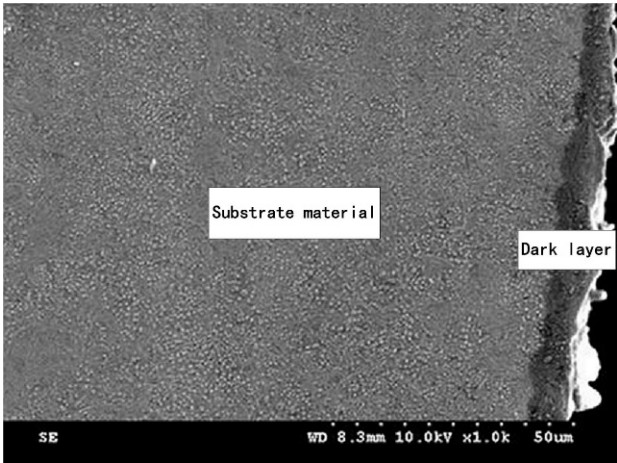

**Figure 10.** Grinding-affected layers (Bearing inner ring cross section).

**Table 8.** Comparison of bearing accuracy after grinding.

|  | SKF Bearing Max | SKF Bearing Min | Max–Min | Local Bearing Max | Local Bearing Min | Max–Min |
|---|---|---|---|---|---|---|
| Roundness of inner ring/μm | 1.1 | 0.9 | 0.2 | 1.8 | 1.2 | 0.6 |
| Roundness of outer ring/μm | 1.5 | 1.3 | 0.2 | 1.9 | 1.4 | 0.5 |
| Inner ring groove/μm | 1.3 | 1 | 0.2 | 1.7 | 1.1 | 0.6 |
| Outer ring groove/μm | 1.2 | 1 | 0.1 | 1.7 | 1.2 | 0.5 |
| Inner ring side swing/μm | 1.7 | 1.5 | 0.2 | 1.9 | 1.5 | 0.6 |
| Outer ring side swing/μm | 1.5 | 1.2 | 0.3 | 2.0 | 1.7 | 0.5 |

This paper developed an ultra-precision rotational accuracy experimental platform of a rolling bearing for analyzing the P4/P2 level super-precision rolling bearing rotational accuracy under working status, and, especially, for analyzing the relationship between the manufacturing imperfection of the bearing components and the rotational accuracy performance, as shown in Figure 11. The principle of this platform is as follows: a high-precision air-bearing spindle is used to support and spin the inner ring, and the radial runout of the outer ring is measured by a high-precision displacement sensor to analyze the rotational accuracy of the test bearing.

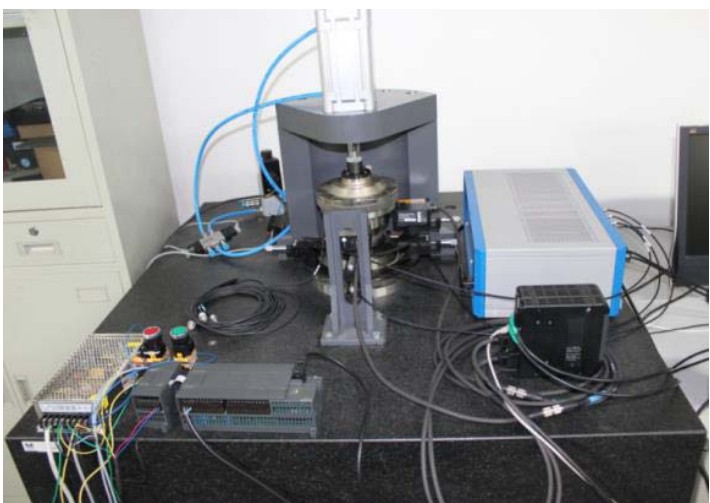

**Figure 11.** Ultra-precision rotational accuracy experimental platform.

## 9. Comparison of Imported and Domestic Bearings

Firstly, the difference between an imported angular-contact ball bearing 7014 and a domestic 7014 was investigated. The measurement compared the imported and domestic 7014 raceway accuracy and rotational accuracy. The gap between the accuracy of the domestic 7014 and the imported 7014 was found. For the bearing raceway's physical properties, the gap between the imported 7014 and the domestic 7014 was analyzed and compared, focusing on several index differences in residual stress, hardness, residual austenite, and grinding metamorphic layer.

Figure 12 compares the imported and domestic bearing grinding deterioration layer. Figure 13 shows the picture of bearing raceway accuracy test.

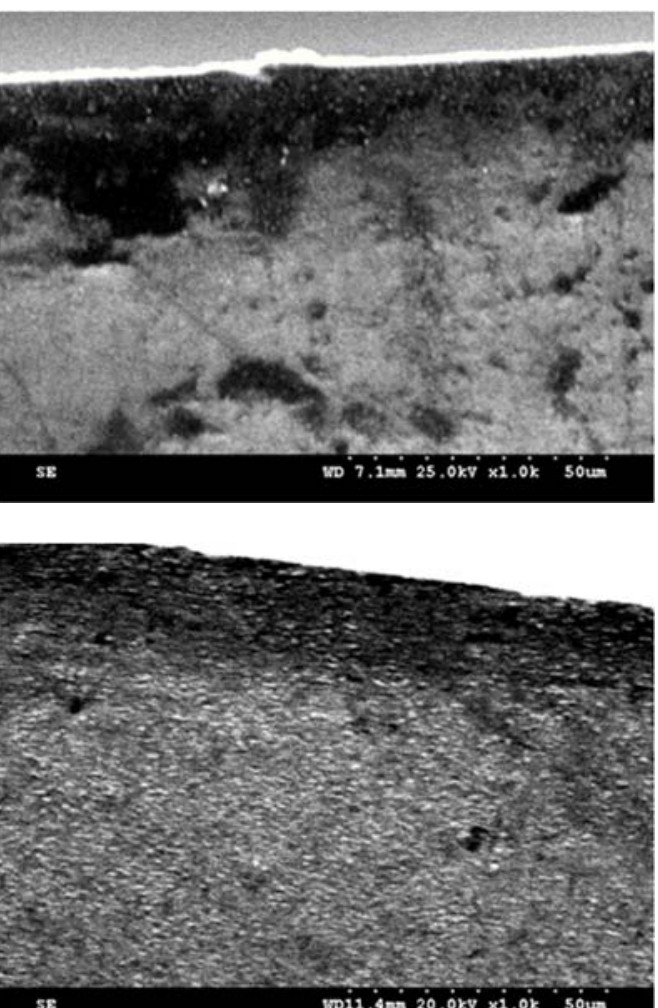

**Figure 12.** Comparison of imported and domestic bearing grinding deterioration layer.

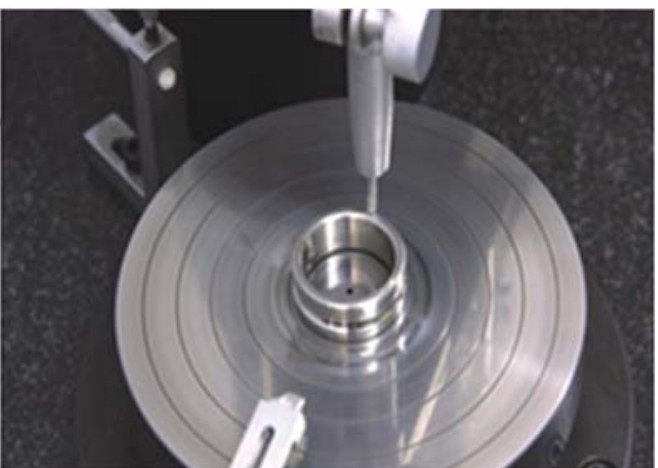

**Figure 13.** Bearing raceway accuracy test.

Figure 14 shows a comparison of burns during the service life of the bearings.

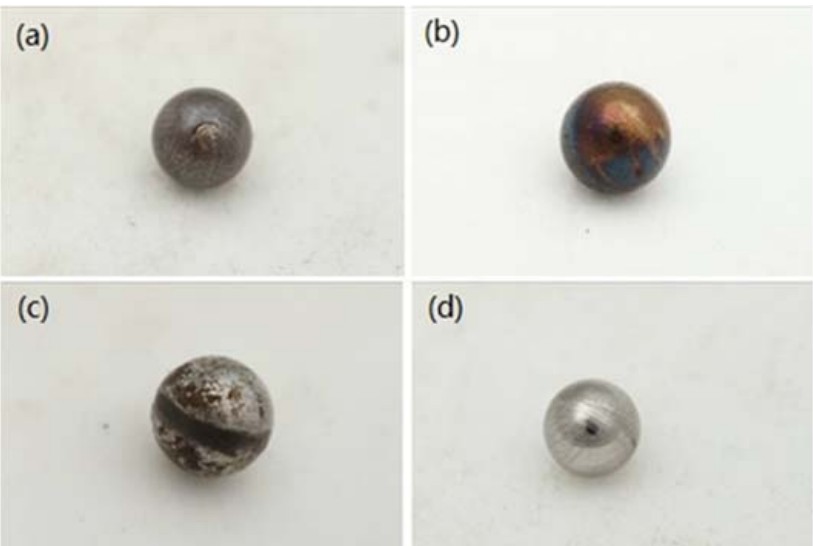

**Figure 14.** Comparison of burns during bearing service. (**a**–**d**) Different forms of steel ball burns.

## 10. Conclusions

(1) Based on the accuracy requirements of the P2-grade rolling bearing raceway, the design indices of the LGID300 bearing raceway grinder were determined. Based on the design indices, the grinding wheel spindle and grinding wheel dressing mode were determined.

(2) According to the design requirements of the functional components, an oil hydrostatic guide rail was designed. A hydrostatic guide rail was used for the X- and Z-axes of the machine tool feed axes. An oil hydrostatic bearing was designed, which was used for the workpiece shaft and turntable of the machine tool. According to the requirements for static pressure, a gap restrictor was designed. The servo drive mode of the torque motor direct drive was designed for the functional components of the guide rail, spindle, and turntable.

(3) The grinding accuracy of the raceway met the design requirements and basically reached P2-grade accuracy.

**Author Contributions:** Z.C. designed the study. L.H. performed the research. Z.C. analyzed the data and wrote the paper. Q.J. made the final revisions and finishing touches to the paper. All authors have read and agreed to the published version of the manuscript.

**Funding:** This research was funded by the Special Funds for Guiding Local Scientific and Technological Development by the Central Government (grant number 22ZY1QA005) and the Gansu Provincial Natural Science Foundation (grant number 21JR11RA066).

**Data Availability Statement:** The raw/processed data required to reproduce these findings cannot be shared at this time as the data also form part of an ongoing study.

**Acknowledgments:** The authors wish to acknowledge Chen from Xi'an Jiaotong University for his help in interpreting the significance of the results of this study.

**Conflicts of Interest:** The authors declare that they have no competing financial interests.

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
