# Peer review of "Design and Development of a High-Speed Precision Internal Grinding Machine and the Associated Grinding Processes"

_processes, doi:10.3390/pr11010064_

Round 1

Reviewer 1 Report

The authors have nicely presented the analysis and the relationship between the manufacturing imperfection of bearing components and the rotational accuracy performance. Further, for analyzing super-precision rolling bearings, an ultra-precision rotational accuracy experimental platform has been used in the present work.

A high positioning accuracy /repetitive-positioning accuracy has been achieved by using suitably controlled hydrostatic bearings.

Author Response

Thank you for your review. The picture of  ultra-precision rotational accuracy experimental platform has been shown in the  paper. 

Reviewer 2 Report

The authors proposed a new grinding machine in order to get the P2 grade. The paper is well structured and describes in detail the parts of the new device with all the improvements. In my opinion, only is necessary minor English corrections and enhancements to the introduction, including in the final paragraph the main results of your approach —congratulation on your contribution. 

Author Response

Thank you for your review. The introduction has been revised. Please see the attachment.

Reviewer 3 Report

This an interesting piece of work and very valuable to the industry - the study investigates using modifications and upgrades to achieve the level of quality required in P2-grade bearings. The work is written well but there a few gaps that need addressing before publication:

At the end of the introduction, you need to connect and introduce all the other sections.

Table 4 swap over mid-table values to the end to compare directly with Table 3.

You mention the comparison, what was the repeatability of the trial to get an idea of standard deviation?

Figure 2 is hard to follow even with someone who has a background in grinding - maybe a front and side view would be better?

Figure 3 is very difficult to see what is going on - the fonts need to be much larger - not sure if a CAD drawing is doing this justice or maybe showing the outline and then home into the areas of significance. Right now very difficult to see what is going on. Also, think about some arrows and text to show significant items to the reader. 

Same with Figure 4 as Figure 3.

Figure 5 is too dark and again difficult to see the details.

in 4.3 should be Renishaw 

Is Figure 11 - the bearing? Please add more information about the geometry being looked at - very hard to picture for the reader.

What would be good would be a general comparison between bearings before modification and upgrades and then after, the modification and upgrades? This would directly show the impact of modifications and upgrades. You could do this via the results of comparisons between the local and SKF bearing. In this comparison, an observation of anomalies would be useful? Anomalies such as burn etc.

With the conclusions have more from the comparison side.

Author Response

#Reviewer:

Table 4 swap over mid-table values to the end to compare directly with Table 3.

Response: Revised.  SKF company bearing manufacturing tolerances and domestic bearing manufacturing tolerances are slightly different. It is worth mentioning that the meaning of the specific parameter values represented in the manufacturing tolerance is also slightly different. But one thing is certain, SKF bearing tolerance variance than the domestic bearing tolerance variance is much smaller.

#Reviewer:

Figure 2 is hard to follow even with someone who has a background in grinding - maybe a front and side view would be better?

Response: Revised.  

The rotary dressing tool is used in this paper to ensure the effect of wheel dressing and reduce the loss of grinding wheels during dressing. The rotary dressing tool is used to finish the dressing of the shaped grinding wheel in this paper by interpolating the workpiece axis, the grinding wheel axis and the rotary table, as shown in Figure 2.

#Reviewer: Figure 3 is very difficult to see what is going on - the fonts need to be much larger - not sure if a CAD drawing is doing this justice or maybe showing the outline and then home into the areas of significance. Right now very difficult to see what is going on. Also, think about some arrows and text to show significant items to the reader. 

Same with Figure 4 as Figure 3.

Figure 5 is too dark and again difficult to see the details.

Response: Revised. Relevant pictures have been modified

#Reviewer: in 4.3 should be Renishaw

Response: Linear scales with German brand Heidenhain

#Reviewer: Is Figure 11 - the bearing? Please add more information about the geometry being looked at - very hard to picture for the reader.

Response: Revised. Relevant pictures were added to the description.

#Reviewer: 

What would be good would be a general comparison between bearings before modification and upgrades and then after, the modification and upgrades? This would directly show the impact of modifications and upgrades. You could do this via the results of comparisons between the local and SKF bearing. In this comparison, an observation of anomalies would be useful? Anomalies such as burn etc.

With the conclusions have more from the comparison side.

Response: Revised. Added comparison images and related instructions
